# Dyeable Hydrophilic Surface Modification for PTFE Substrates by Surface Fluorination

**DOI:** 10.3390/membranes13010057

**Published:** 2023-01-02

**Authors:** Mizuki Kobayashi, Fumihiro Nishimura, Jae-Ho Kim, Susumu Yonezawa

**Affiliations:** 1Department of Materials Science and Engineering, Faculty of Engineering, University of Fukui, 3-9-1 Bunkyo, Fukui 910-8507, Japan; 2Cooperative Research Center, University of Fukui, 3-9-1 Bunkyo, Fukui 910-8507, Japan

**Keywords:** polytetrafluoroethylene, Au sputtering, surface fluorination, dyeable layer

## Abstract

Polytetrafluoroethylene (PTFE) is the most widely used fluoropolymer that has various functionalities such as heat resistance, chemical resistance, abrasion resistance, and non-adhesiveness. However, PTFE is difficult to dye because of its high water repellency. In this study, the PTFE surface was modified by a combination of gold sputtering and surface fluorination to improve dyeability. X-ray photoelectron spectroscopy indicated that, compared with the untreated sample, the gold-sputtered and acid-washed surface of PTFE had a negligible number of C–F terminals. Furthermore, the intensity of the C–C peak increased drastically. The polar groups (C=O and C–F_x_) increased after surface fluorination, which enhanced the electronegativity of the surface according to the zeta potential results. Dyeing tests with methylene blue basic dye showed that the dye staining intensity on the surface of fluorinated PTFE samples was superior to other samples. It is due to the increased surface roughness and the negatively charged surface of fluorinated PTFE samples. The modified PTFE substrates may find broad applicability for dyeing, hydrophilic membrane filters, and other adsorption needs.

## 1. Introduction

Polytetrafluoroethylene (PTFE) has unparalleled heat and chemical resistance among fluororesins, as well as non-adhesiveness and electrical insulation properties. Because it has various functions, it is a fluororesin that is widely used industrially in the chemical and medical fields [1,2,3,4,5,6,7]. In particular, PTFE has been used as an attractive membrane material owing to its superior chemical resistance, good thermal stability, and high mechanical strength, which makes it widely used in environmental protection, filtration, textiles, medicine, military, etc. [8,9,10,11,12,13,14,15,16,17,18]. However, PTFE exhibits both hydrophobic and oleophobic properties owing to its low surface energy. Furthermore, because PTFE has a strong C–F bond, it does not show significant interaction with some compounds, such as organic dyes. Organic dyes are helpful for evaluating the adsorption onto PTFE in general. In dye-related industries, water pollution by organic dyes is regarded as a serious problem. To better understand this problem, the adsorption of organic dyes onto solid media such as PTFE has been investigated for a long time [19,20]. However, the hydrophobic nature of the membrane prevents the penetration of aqueous solutions into the pores, while the pore size and shape determine the diffusion or convection across the membrane. Therefore, hydrophilic modifications of PTFE are beneficial for dyeing, metal plating, and other coating needs [21,22,23,24,25]. Common surface modification methods, such as sodium naphthalene chemical treatment, high-temperature sintering, and irradiation grafting, have been carried out [26,27]. However, these methods can readily damage PTFE structures and cause environmental pollution. As a dry treatment method, plasma treatment is widely used for the surface modification of PTFE, which can introduce a variety of active functional groups on the surface in a short time [28,29,30,31,32,33]. However, the wettability after plasma treatment depends on numerous process parameters, such as the type of discharge, feed gas, working pressure, input power, and treatment time. Furthermore, this method cannot be used for complex geometric shapes.

In a previous study [34,35,36], the surfaces of polyethylene terephthalate (PET) and polycarbonate (PC) were modified by direct fluorination to achieve strong adhesion with the plating (i.e., good dyeability). The dyeing properties of the fluorinated samples were enhanced by the increased roughness and hydrophilicity. The process of direct fluorination involves a gas-phase chemical reaction between fluorine gas (F_2_) and the polymer surface. This is an effective chemical method for modifying and controlling the physicochemical surface properties of polymers. Because it’s harder to break C-F in perfluorinated samples than C-H in nonperfluorinated samples under the same conditions. In addition, the surface of polymeric membranes must include adequate pore size, roughness, and wettability for the adsorption of organic dyes. Ohkubo et al. reported that PTFE can be defluorinated using an alkali metal amalgam to produce porous carbon with a large surface area [37,38,39].

In this study, we report on the effects of gold sputtering and surface fluorination on the dyeing properties of PTFE substrates. The adsorption of organic dyes on PTFE may be used to remove organic dyes, which are toxic substances in the environment. 

## 2. Materials and Methods

### 2.1. Surface Modification of PTFE

The entire preparation process for fluorinated PTFE is shown schematically in Figure 1. PTFE sheets (NAFLON) were obtained from the NICHIAS Corporation, Tokyo, Japan. PTFE plates (10 × 10 × 1.0 mm) were washed with ethanol to remove organic residues from the surface. Gold-sputter deposition on PTFE plates was performed using an ion coater (IC-50, Shimadzu Manufacturing, Kyoto, Japan). The target (99.99% Au) was fixed horizontally at the top-center of the deposition chamber. The glow discharge was allowed under low vacuum (less than 20 Pa). The discharge conditions were 6 mA and 1.4 kV for a sputtering time of 3 min. To dissolve the sputtered gold layers on PTFE, the sample was soaked in an aqueous solution for 1 min. It was then washed and dried at 40 °C for 24 h. Surface fluorination of the Au-coated and washed PTFE samples was carried out using F_2_ gas. F_2_ (99.5% purity) was produced by electrolysis of a KF/HF mixture in HF solution. The PTFE plates were placed in a nickel reactor (24 mm × 32 mm × 5 mm) and held at 25 °C under vacuum (0.1 Pa) for more than 10 h to eliminate impurities before use. The fluorination apparatus was explained in a previous paper [40], which used a reaction temperature of 25 °C, a gas pressure of 13–101 kPa, and a reaction time of 1 h. The sample names and reaction conditions are listed in Table 1.

### 2.2. Material Characterization

The surface morphologies of the various PTFE samples were observed using scanning electron microscopy (SEM, S-2400; Hitachi Ltd., Tokyo, Japan). The surface topography was evaluated using atomic force microscopy (AFM; Nanoscope IIIa, Digital Instruments, Inc., Tokyo, Japan). Scanning was performed in the tapping mode within an area of 5 × 5 µm^2^. The arithmetic mean surface roughness (*Ra*) was determined using the AFM roughness profile. The static water contact angles of the untreated and modified PTFE were measured at 25 °C using the sessile drop method. A 10 μL water droplet was used in a telescopic goniometer with a magnification power of 23× and a protractor with a graduation of 1° (Krüss G10, Hamburg, Germany). Five measurements were obtained at different surface locations on each sample to determine the average value (±2°). The surface chemical states of the untreated and modified PTFE were determined using XPS (JPS-9010, JOEL, Tokyo, Japan). All the binding energies were referenced to a carbon peak at 284.5 eV. The chemical compositions of the untreated and modified PTFE samples were examined by Fourier-transform infrared (FTIR) absorption spectroscopy (Nicolet 6700; Thermo Electron Scientific, Waltham, MA, USA). The analysis was performed in transmittance mode within the range of 500–4000 cm^−1^, where 32 scans were obtained, and air background removal was conducted. The zeta potential data of PTFE samples were measured using a solid sample cell unit with a zeta potential device (ElSZ-2; Otsuka Electronics Co. Ltd., Osaka, Japan).

### 2.3. Dye Staining of PTFE

Methylene blue (MB; Fujifilm Wako Chemical Corp., Hirono, Fukushima, Japan) and Aced Red 52 (AR52; Fujifilm Wako Chemical Corp., Hirono, Fukushima, Japan) were used as the representative basic and acidic dyes, respectively. Staining solutions containing 0.01 mol/L of dye in ultrapure water were set in a water bath at 80 °C, and the PTFE samples were immersed in the staining solutions for 30 min. The PTFE samples were subsequently washed with ultrapure water and dried in air. The surface staining of each sample was evaluated based on the N and S contents determined using XPS. The stability of the color layer on the PTFE samples was evaluated after sonication in water at 80 kHz for 1 h using an ultrasonic multicleaner (W-115, Honda Electronics Co., Ltd., Toyohashi, Japan).

## 3. Results and Discussions

### 3.1. Surface Modification of PTFE Plates Using Gold Sputtering and Fluorine Gas

Figure 2 shows the photographs and XPS (Au4f) results of the Au-sputtered samples before and after washing with aqua regia solutions. After Au sputtering, a metal layer was observed in the Au-P samples. The Au 4f spectra in the XPS results show that the sputtering layer was Au metal. After washing with the aqua regia solution, the Au layer was completely removed, as indicated in Figure 2 (Au-W-P).

The surface morphology and roughness of the PTFE samples were characterized using SEM and AFM, respectively, as shown in Figure 3. SEM and AFM images indicated that the untreated PTFE plates had quite smooth and uniform surface with a low surface roughness of ~8.1 nm (Figure 3). However, the surface morphology clearly changed after Au sputtering and washing, as shown in the FE-SEM image of the Au-W-P samples. The surface of the Au-W-P samples had mass buildups or irregular fine cracks and voids, which were similar to netty pores. This is likely due to the formation of byproducts with fluoride after Au sputtering, which were subsequently washed away by the aqua regia solution. When fluoride is removed, some space is created in the form of pores in the carbon matrix. The surface roughness of the Au-W-P samples increased to 18.7 nm, as indicated by the AFM results (Figure 3). Furthermore, after surface fluorination, fine irregularities significantly increased on the surface of the Au-W-13F-P samples and the surface exhibited valley-shaped morphology. Some rounded products were also observed on the surface, which may be agglomerations of molecular segments created in the process of Au sputtering and acid washing. The surface roughness of the Au-W-13F-P samples was approximately ten times greater than that of the untreated PTFE sample (i.e., 8.1 nm, indicated by the AFM results (Figure 3). When the fluorine pressure was increased, the fluoride layer expanded on the PTFE surface. This was due to a decrease in the surface roughness of the Au-W-101F-P samples.

Wettability, which was measured using contact angle measurements, is an important factor for polymer membranes. The water contact angle of the untreated PTFE sample was 107°, as shown in Figure 4, whereas those of the surface-modified PTFE samples were lower owing to the increased surface roughness (Figure 3). However, the water contact angle of Au-W-101F-P sample (50°) was lower than that of Au-W-13F-P, despite its lower surface roughness. The roughness of the PTFE surface increased after Au sputtering and surface fluorination. Some hydrophilic groups were also generated on the PTFE surface, which increased the polarity and surface energy; thus, the hydrophilicity of the PTFE surface improved significantly. This may be attributed mainly to the defluorination of hydrophobic groups, such as –CF_2_ on PTFE, due to the gold sputtering and acid washing process.

The F 1s, O 1s, and C 1s XPS spectra of the untreated and modified PTFE samples are shown on Figure 5. The F 1s spectra of the untreated PTFE samples indicated strong –CF_2_ bonding at 689.8 eV, which was almost eliminated after Au sputtering and washing. After surface fluorination, the main peak was found at 687 eV, which was assigned to –C–Fx peaks. The intensity of the O 1s peak increased significantly after the Au sputtering and washing, while it shifted to a higher binding energy after the surface fluorination, as shown in Figure 5. The elemental compositions of the PTFE samples were also evaluated from the XPS results (Figure 5), and are presented in Table 2. Notably, the oxygen content of Au-W-P sample was approximately five times larger than that of the untreated PTFE (4.93%). Furthermore, the fluorine content of the fluorinated samples (i.e., 49.04% and 47.16%) increased by more than two times that of the Au-W-P sample (i.e., 21.80%). The higher shift of the O 1s peak in the fluorinated samples is caused by the formation of fluorides such as –COF_x_. Evidently, the untreated PTFE shows peaks at 284.4 and 292.4 eV, which are ascribed to C–C and C–F bonds in PTFE as shown in the C 1s results (Figure 5). However, after the gold sputtering and washing process for the Au-W-P sample, the C–F peak at 294.4 eV nearly disappeared while the C–C peak at 284.4 eV increased drastically. From the results of peak separation with Gaussian distributions (dotted line), a new peak was also found at 286.0 eV, which was assigned to C–O or C–OH bonds. This means that carbon radicals originating from the defluorination of PTFE reacted with other carbon radicals and water in the acid solution, thereby resulting in the formation of C–C cross-links [41] and hydrophilic surfaces. Moreover, after the surface fluorination, these C–C and C–O bonds changed to either –C=O or O=C–OH, and –C–F_x_ at 288.4 eV [42]. Notably, the formation of both polar groups (–C=O and –C–Fx) enhanced the preparation of hydrophilic PTFE surfaces. The partial polarity of the surface was improved by the addition of F, as its high electronegativity and acidity easily attracted water as a polar solvent.

As shown in Table 2, the F/C elemental composition ratio (2.26) of the untreated PTFE sample decreased drastically (to 0.41) after the gold sputtering and washing process. However, the F/C ratio could again be increased (to 1.77) by the surface fluorination because of the introduction of fluorinated bonds such as C–F_x_ or C–OF. The effects of the change in F_2_ pressure on the F/C ratio on the PTFE surface appear to be insignificant. For example, increasing the pressure to 101 kPa causes the F/C ratio to decrease slightly to 1.64. This may be attributed to the formation of CF_4_ or COF_2_ gas on the surface of the Au-W-101F-P sample.

The FTIR spectra of the untreated and modified PTFE samples are shown on Figure 6. The spectrum of the untreated PTFE sample exhibited absorption bands at 1203 and 1147 cm^−1^ (corresponding to the –CF_2_ stretching vibration peak). The absorption bands of the other modified PTFE samples appeared to be the same as those of the untreated samples. To verify the effects of surface modification on the surface structure of PTFE, the vertical axis of the FT-IR spectra was enlarged, as shown on Figure 6 (bottom). The new peak at 1646 cm^−1^ corresponded to the C=C resonance vibrations in the cross-links of the Au-W-P samples. After the surface fluorination, the intensities of the absorption bands at 1726 cm^−1^ (C=O stretching vibration peak) and 1852 cm^−1^ (O–F stretching vibration peak) increased. Moreover, after the gold sputtering and washing process, a broad peak appeared in the range of 3500–3100 cm^−1^, corresponding to the OH stretching vibration peaks. This broad peak was intensified after the surface fluorination at high F_2_ pressure. With increasing surface fluorination, the signals indicated a surface rearrangement and a reaction with hydroxyl groups that were dissociated from the moisture in air, which bonded to the carbon radicals due to CF_4_ gasification. This strong hydrophilic bond led to a decrease in the contact angle with water, as shown in Figure 4.

After surface fluorination, the fluorinated bonds (C–Fx or C–OF) as shown in XPS results increased on the PTFE surface. Additionally, these fluorinated bonds indicated high electronegativity from the results of zeta potential (Figure 7). The Zeta potential (−40.41 mV) of Au-W-13F-P samples was about 5 times negatively larger than that of untreated sample. This result is similar to the results in previous studies [35,36].

### 3.2. Dyeing of Surface-Modified PTFE Plates

Dyeing tests were performed using MB and AR52 solutions as the representative basic and acidic dyes, respectively. Dye staining of the untreated and modified PTFE samples was performed using MB solutions (Figure 8). No staining was observed in the untreated samples using the MB solutions, but the Au-W-P samples were stained with MB solutions. An increasingly deeper color was achieved by surface fluorination, as shown for the Au-W-13F-P and Au-W-101F-P samples.

The exhaustion of the MB dye after surface staining of the PTFE samples was evaluated by XPS analysis (Figure 9). The chemical formula of MB is C_16_H_18_ClN_3_S, and the N and S contents of the adsorbed MB were determined from the N 1s and S 2p3/2 XPS spectra. In the untreated samples, N 1s and S 2p3/2 peaks were not detected. However, both N 1s and S 2p3/2 peaks were observed for the Au-W-P samples. Moreover, the intensities of the N 1s and S 2p3/2 peaks of the fluorinated samples were much higher than those of the Au-W-P samples. The intensities of N 1s and S 2p3/2 peaks increased with increasing F_2_ pressure in fluorinated samples. The intensities of the N 1s and S 2p3/2 peaks were proportional to the visibly observed degree of deep coloring (Figure 8). The dyeability of the fluorinated samples was attributed to their higher roughness and higher electronegativity (Figure 7). MB, being a salt with a cationic component, underwent an easy adsorption on the negative surface of the fluorinated PTFE samples via Coulomb attraction. 

In the case of staining with the acidic AR52 dye, only slight staining of the fluorinated PTFE samples had occurred, as shown on Figure 10c. Thus, fluorinated PTFE can be effectively stained using basic dyes (Figure 10b), but not using acidic dyes (Figure 10c). This may be attributed to the physical adsorption via Coulomb attraction between the positively charged dyes and negatively charged PTFE surface. The negative charge on the PTFE surface may be induced by the polar groups (i.e., –C=O and –C–F_x_) created by surface fluorination, as indicated by the XPS results (Figure 5). In the case of aqua regia washing without gold sputtering, no staining of the PTFE samples (W-P) had occurred (Figure 10a). This indicates that defluorination using gold sputtering is important for the surface activation of PTFE plates. Surface fluorination can create a dyeable negative surface on the PTFE plates. This dyeable hydrophilic surface on the modified PTFE can be kept after a few washings with water.

To confirm the chemical resistance of modified surface, the Au-W-13F-P samples were wholly immersed in HCl solution (1 mol/L) and KOH solution (2 mol/L) as the representative strong acidic (pH 1.0) and alkaline solutions (pH 13), respectively, for 10 min. Dye staining of surface treated Au-W-13F-P samples was performed using MB solutions, as indicated in Figure 11. The good wettability of Au-W-13F-P samples can be kept and slightly improved after treatment with (b) acidic solution and (c) alkaline solution, as shown in the results of water contact angles. Comparing with (a) untreated sample, the dyeability of (b) acid treated samples and (c) alkali treated samples was almost the same. It means that the high chemical resistance of modified PTFE surface can be still kept even after treatment with strong acidic and alkaline solutions. Thus, the dyeable hydrophilic surface on the modified PTFE may considerably expand the range of applications of PTFE as membrane filters.

## 4. Conclusions

Hydrophobic PTFE plates were successfully modified via step-by-step gold sputtering and surface fluorination. All the modified PTFE plates showed improved hydrophilicity compared to the untreated PTFE. After gold sputtering and washing with an aqua regia solution, the surface roughness increased, and the wettability improved. This led to defluorination and subsequent cross-linking on the PTFE surface, as indicated by the XPS results. Surface fluorination may drastically increase the roughness and hydrophilicity of PTFE, as shown by the AFM results and water contact angle tests. XPS and FTIR results indicated the formation of polar (–C=O and –C–F_x_) and hydroxyl groups on the surface of PTFE. The fluorinated PTFE surface exhibited superior dye staining with the basic MB dye, but not with the acidic AR52 dye. This indicated that fluorinated PTFE surfaces may be effectively stained using basic dyes, but not using acidic dyes. The chemical resistance of modified PTFE surface can be highly kept even after treatment with strong acidic and alkaline solutions. Consequently, the surface dyability of PTFE was enhanced by the combination of gold sputtering and surface fluorination, which is owed to the increased surface roughness and induced negatively charged hydrophilic surface.

## Figures and Tables

**Figure 1 membranes-13-00057-f001:**
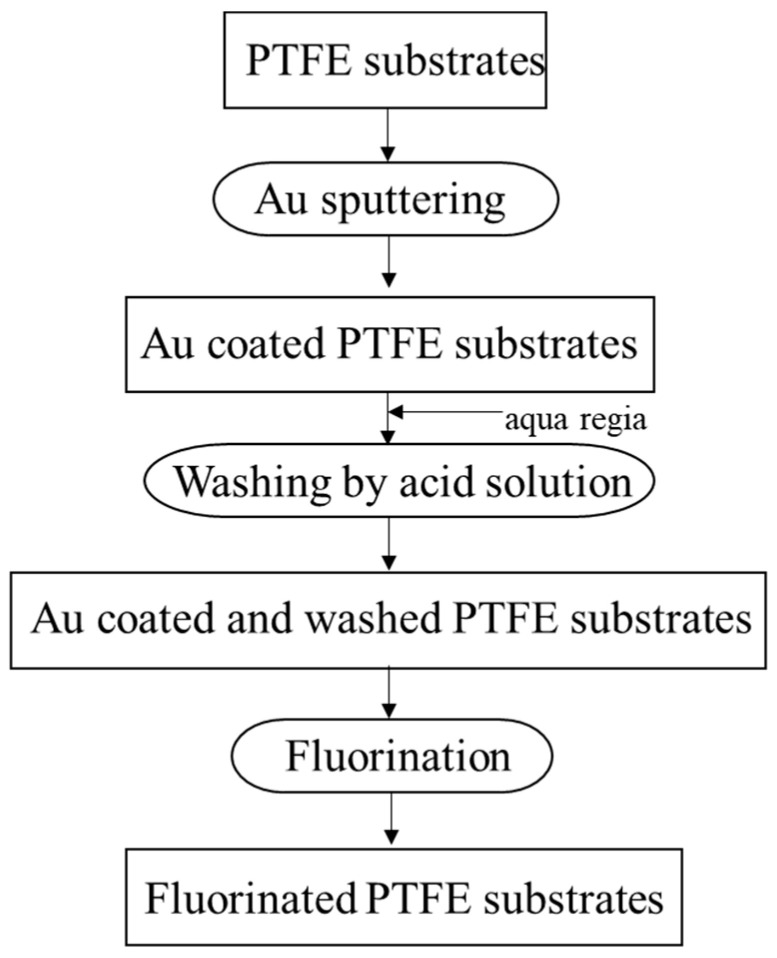
Flow diagram for preparation of fluorinated PTFE.

**Figure 2 membranes-13-00057-f002:**
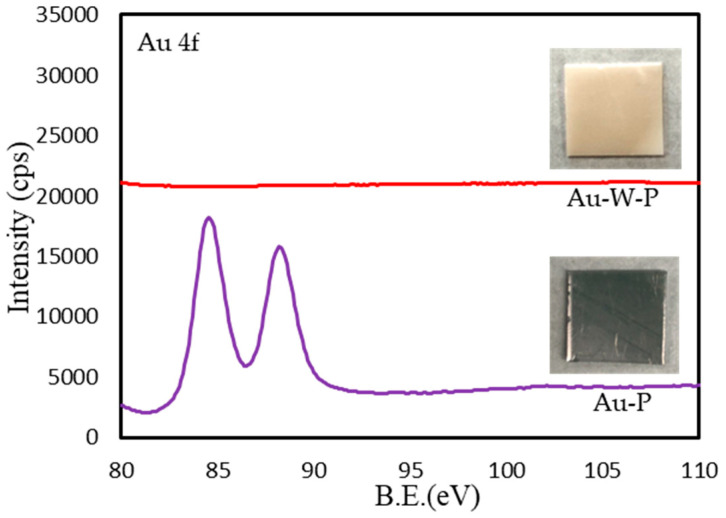
Photographs and XPS (Au 4f) results of Au sputtering samples before and after washing with aqua regia solutions.

**Figure 3 membranes-13-00057-f003:**
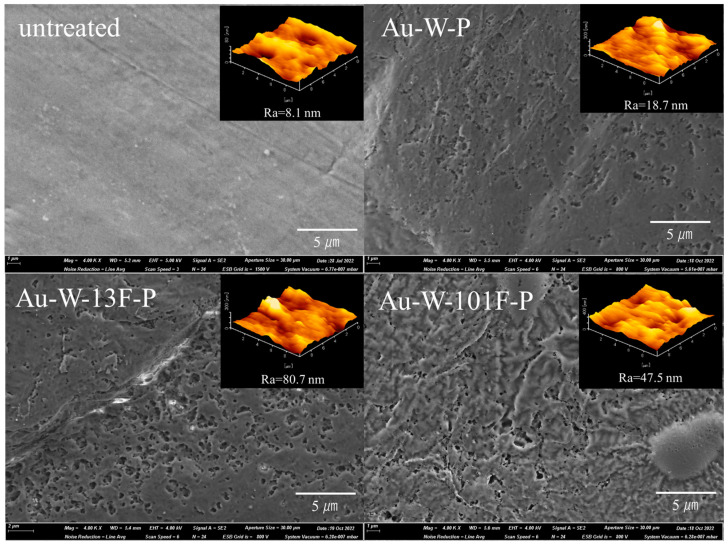
SEM images and AFM images of various PTFE samples.

**Figure 4 membranes-13-00057-f004:**
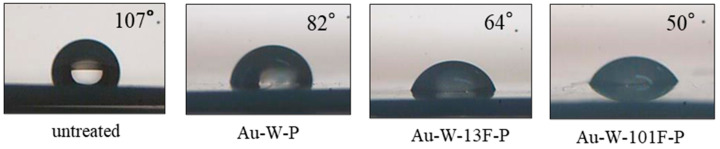
Contact angles of various PTFE samples using water.

**Figure 5 membranes-13-00057-f005:**
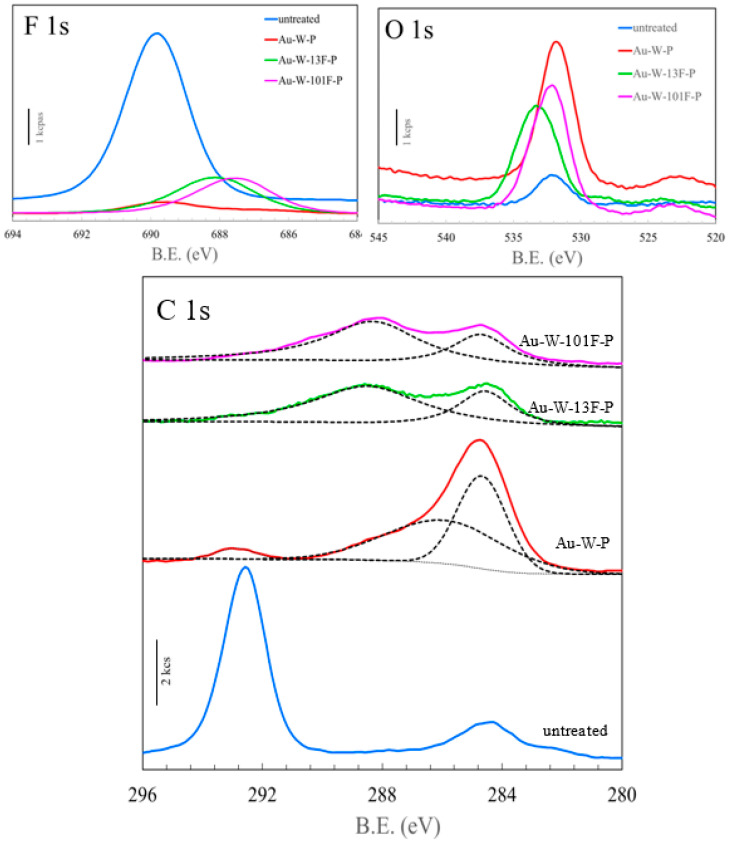
XPS spectra of untreated and modified PTFE samples.

**Figure 6 membranes-13-00057-f006:**
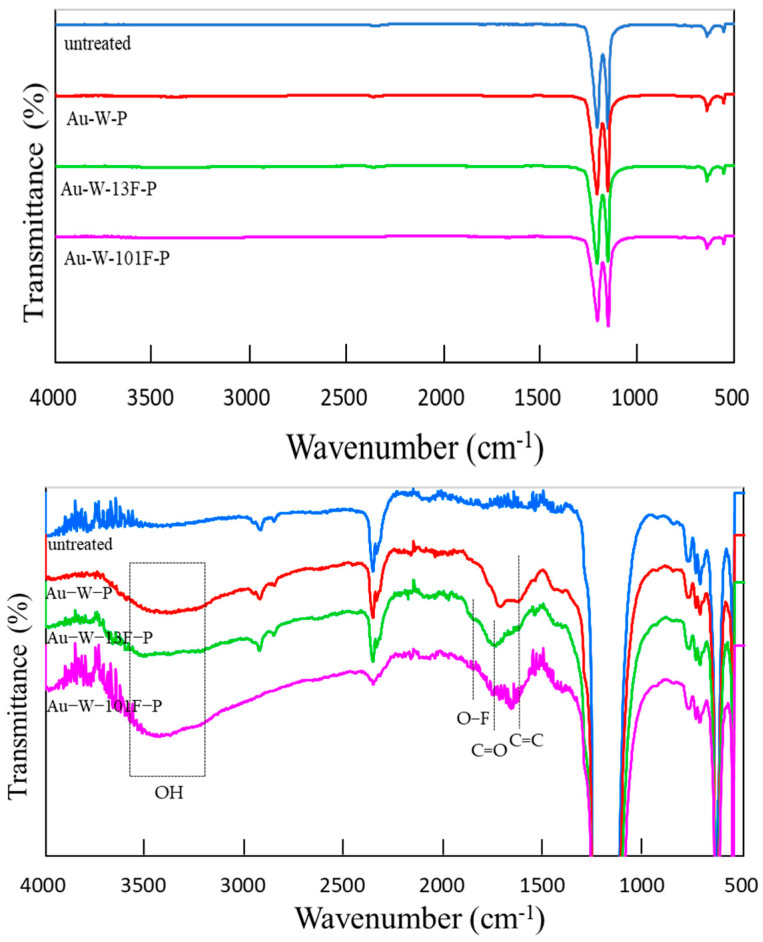
FTIR spectra of various PTFE samples.

**Figure 7 membranes-13-00057-f007:**
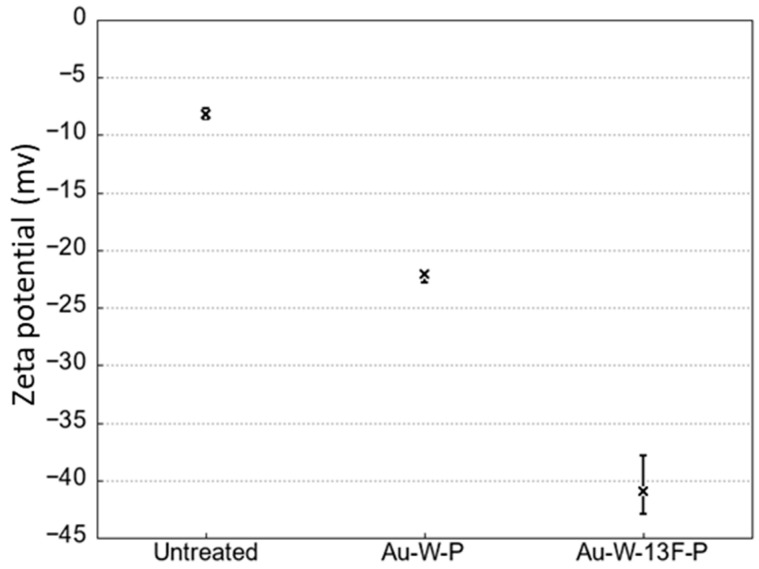
Zeta potential of untreated and modified PTFE samples with water at a constant pH 7.0.

**Figure 8 membranes-13-00057-f008:**
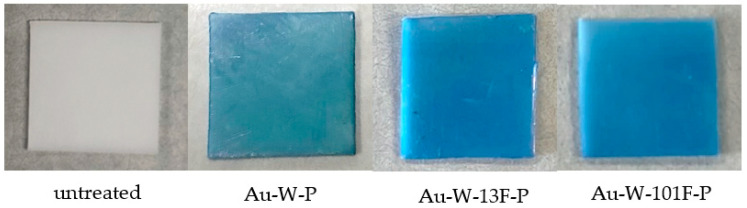
Photographs of dye staining of various PTFE samples with methylene blue.

**Figure 9 membranes-13-00057-f009:**
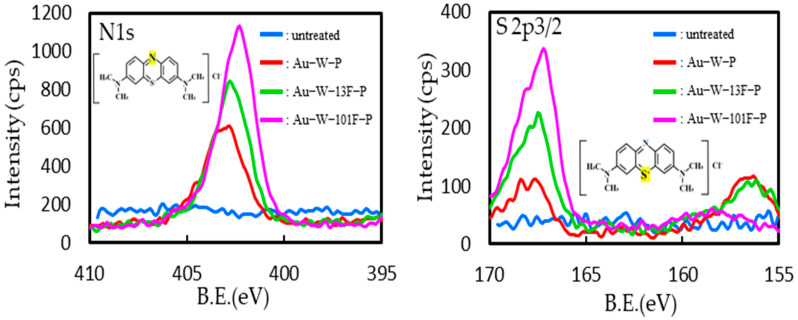
XPS (N 1s and S 2p3/2) spectra of various PTFE samples dyed with methylene blue solution.

**Figure 10 membranes-13-00057-f010:**
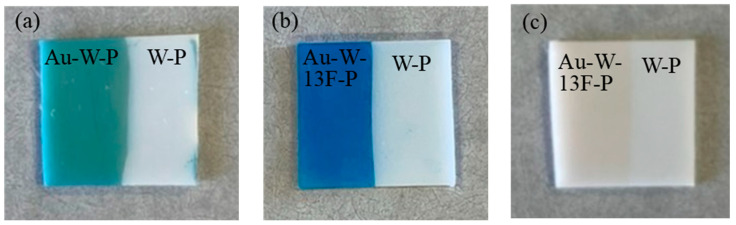
Photographs of dye staining of various PTFE samples with (**a**,**b**) methylene blue and (**c**) AR52 solutions.

**Figure 11 membranes-13-00057-f011:**
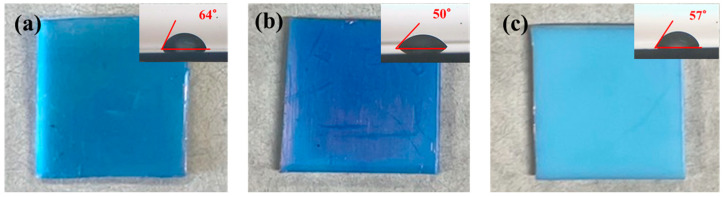
Photographs of dye staining and water contact angles of (**a**) untreated, (**b**) acidic treat-ed, and (**c**) alkali treated Au-W-13F-P sampes.

**Table 1 membranes-13-00057-t001:** Sample names and reaction conditions.

Sample Name	Au Coating	Washing by Aqua Regia	Surface Fluorination		
			Temp. (°C)	Time (h)	Pressure (kPa)
untreated	-	-	-	-	-
Au-P	with	without	-	-	-
Au-W-P	with	with	-	-	-
Au-W-13F-P	with	with	25	1	13
Au-W-101F-P	with	with			101

**Table 2 membranes-13-00057-t002:** Surface elemental composition of PTFE samples evaluated from XPS results (Figure 5).

Sample Name	Surface Composition Ratio (%)			F/C
	C	O	F	
untreated	29.19	4.93	65.87	2.26
Au-W-P	52.67	25.53	21.8	0.41
Au-W-13F-P	27.73	23.23	49.04	1.77
Au-W-101F-P	28.67	24.18	47.16	1.64

## Data Availability

The data presented in this study are included in this article.

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
