# Peer review of "Dyeable Hydrophilic Surface Modification for PTFE Substrates by Surface Fluorination"

_membranes, 2023, doi:10.3390/membranes13010057_

Round 1
Reviewer 1 Report
I recommend Major Revision
1. Please add the optimum results in the abstract and replace the explanation like words [for example…] to discussions.
2. The author should make a comparison table between their work and other previous work, especially in substrate performance.
3. What are the novelty and motivation of this work?
4. The results should be enhanced by the absorption kinetic model.
5. Most results need more discussions.
6. What is the benefit of using Au, especially it is considered the high cost and increases the cost of the enhanced support?
Author Response
"Please see the attachment."

Reviewer 2 Report
In this study, the authors proposed a simple hydrophillic modification to hydrophobic PTFE sheets to allow for dye adsorption. The idea of such modification is not novel. However, the application of dye adsorption is interesting. The authors can emphasize more on the dye adsorption application. Furthermore, since the authors mentioned waste water containing dyes, they should be testing the efficiency of their modified PTFE sheet on separating dyes with harsher chemicals that cannot be separated using other materials. This will greatly strengthen their article. I suggest publishing with major revision on the experiments to explain the need of such simple modification and prove that after the modification the PTFE can still be chemical resistant.
Author Response
"Please see the attachment."

Round 2
Reviewer 1 Report
The author did all the modifications, I recommend the acceptance
Author Response
Dear Reviewer
Thank you for your advice and comments regarding our paper (Dyeable hydrophilic surface modification for PTFE substrates by surface fluorination).
Reviewer 2 Report
The authors did not add any significant experiments. As a reader, I still do not see the significant need to make the hydrophobic membrane hydrophillic, as there are no evidence as to whether it is even chemical resistant after the modification.
Author Response
"Please see the attachment."
